The tip of the iceberg: extraordinarily high diversity while examining two infralittoral nematode communities on Okinawa-jima Island, Japan, using morphology and DNA barcoding

Carletti Marilyn 1
Viñuela Rodríguez Nuria 1
Rossetti Gaia 1 2
Rossi Virginia 1 2
Tan Bryan Gabriel Pulido 3
Reimer James Davis jreimer@sci.u-ryukyu.ac.jp 1 4
1 Molecular Invertebrate Systematics and Ecology Lab, Graduate School of Engineering and Science, University of the Ryukyus , Nishihara , Okinawa , Japan
2 Marche Polytechnic University , Ancona , Italy
3 University of the Ryukyus , Graduate School of Engineering and Science, Nishihara , Okinawa , Japan
4 Tropical Biosphere Research Center, Graduate School of Engineering and Science, University of the Ryukyus , Nishihara , Okinawa , Japan
.Zyła Dagmara
Electronic publication date: 2025 Jul 30
Publication date: 2025
Volume: 13
Electronic Location ID: e19757
Received 2025 Mar 3; Accepted 2025 Jun 25
Copyright: ©2025 Carletti et al.
Copyright year: 2025
Copyright holder: Carletti et al.
License: This is an open access article distributed under the terms of the Creative Commons Attribution License, which permits unrestricted use, distribution, reproduction and adaptation in any medium and for any purpose provided that it is properly attributed. For attribution, the original author(s), title, publication source (PeerJ) and either DOI or URL of the article must be cited.
License URL: https://creativecommons.org/licenses/by/4.0/

Keywords: Infralittoral, Biodiversity, Coral reefs, Ryukyu archipelago, Roundworms

Funding: Japan Science and Technology’s “COI-Next” project hosted by the Okinawa Institute of Science and Technology (OIST) The Ministry of Education, Culture, Sports, Science and Technology (MEXT - Monbukagakusho) James Davis Reimer received financial support from Japan Science and Technology’s “COI-Next” project hosted by the Okinawa Institute of Science and Technology (OIST). Marilyn Carletti received a scholarship from the Ministry of Education, Culture, Sports, Science and Technology (MEXT - Monbukagakusho) to support her Ph.D. The funders had no role in study design, data collection and analysis, decision to publish, or preparation of the manuscript.

==============================
Background

Nematodes are among the most diverse and abundant metazoans in aquatic habitats, contributing significantly to global biodiversity. Despite their abundance and importance, the presumed number of undescribed species is high and their diversity is often underestimated.

Methods

In this research, sediment samples were collected from three microhabitats (bare sand, seagrass, coral) in two sites around Okinawa-jima Island in subtropical southern Japan. Nematode specimens were obtained by filtering the sediment and were then used to determine meiofaunal assemblages with morphology and molecular methods at the two sites and to compare them with environmental variables.

Results

The results showed an overwhelmingly high biodiversity of nematofauna with both methods. The morphological identification of free-living nematodes was partly supported by molecular analyses, with the results varying more regarding less common taxa. The discrepancies between different methods may be due to low success of DNA amplifications, high nucleotide variability, and overestimation of congeneric specimens. We observed that coral reef habitats clearly differed from nearby sand and seagrass beds in terms of nematode genus-level assemblages. We identified at least 10 orders and 38 genera of nematodes from our samples that only span two different sites, and it is highly likely these samples include undescribed taxa. Our results strongly suggest that coral reefs and neighboring areas are hot-spots for nematode diversity, at least around Okinawa-jima Island if not also in other coral reef regions.

Introduction

Free-living nematodes are among the most diverse and abundant metazoans in aquatic habitats (Appeltans et al., 2012; Lambshead, 2004). Nematodes contribute substantially to worldwide biodiversity, especially in marine sediments, where they may reach very high abundances (Grassi et al., 2022; Moens et al., 2013; Semprucci et al., 2018).

Nematode identification is particularly challenging and it is usually possible only by trained taxonomic experts (Bogale, Baniya & Di Gennaro, 2020), which are unfortunately reported as decreasing in number (Nisa, Tantray & Shah, 2022). Because of the presumed high number of undescribed nematode species and the overall lack of current research, participation from students and researchers is limited (Appeltans et al., 2012; Ridall & Ingels, 2021). With the decline of the number of taxonomists (Boufahja et al., 2010; Ridall & Ingels, 2021; Nisa, Tantray & Shah, 2022) and the concurrent rise in molecular methods, DNA barcoding now plays a significant role in nematode diversity research. Most nematode DNA barcoding studies initially used the cytochrome c oxidase sub-unit 1 (COI) gene fragment because of its ubiquity and suitability for many organisms. The COI gene has been successfully used in the barcoding of marine nematodes to resolve taxonomic relationships among closely related and/or cryptic species (Blouin et al., 1998; Derycke et al., 2005). However, this marker has been shown to have low amplification success overall for nematodes as a result of mutations in primer-binding regions (Blaxter et al., 1998; De Ley et al., 2005; De Ley, 2006). More recent barcoding studies (Armenteros et al., 2014; Avó et al., 2017; Bik et al., 2010a; Bik et al., 2010b; Chariton et al., 2014; Derycke et al., 2010; Pereira et al., 2010; Schenk, Kleinbölting & Traunspurger, 2020) that investigated meiofaunal organisms including nematodes have suggested the use of nuclear 18S small subunit (SSU) ribosomal DNA (18S rDNA) and 28S large subunit (LSU) ribosomal DNA (28S rDNA). Even though no standardized gene for DNA barcoding is available for marine nematodes, the accumulation of nematode 18S and 28S nuclear ribosomal DNA (rDNA) sequences in public databases reflects their utility and widespread use in molecular phylogenetic studies (Avó et al., 2017; Blaxter et al., 1998; Kumar, Sen & Bhadury, 2014; Litvaitis et al., 2000; Nadler, 1992; Nadler et al., 2007; Pereira et al., 2010).

Okinawa-jima Island is the largest island of the Ryukyu Archipelago, which spans the region between southern mainland Japan to Taiwan. Okinawa-jima Island is situated in an area with high marine biodiversity that is under threat from numerous local anthropogenic impacts (Roberts et al., 2002) including terrestrial runoff (Shilla et al., 2013), pollution (Takeuchi, 2023), and coastal development (Masucci & Reimer, 2019). Despite these threats, marine biodiversity research in the surroundings of Okinawa-jima Island, as in many places in the world, remains unbalanced, with a clear need for more research on so-called “minor” taxa such as nematodes, as well as more research on marine ecology and conservation also being critically needed (Reimer et al., 2019).

Until now, the diversity of marine nematodes of Okinawa-jima Island has been little studied and only a few reports are available (e.g., Leduc & Sinniger, 2018; Leduc, 2022). The aim of this study was to help address this research gap, and we assessed the marine diversity of the phylum Nematoda from two shallow coral reef sites around Okinawa-jima Island, Japan, by comparing the results obtained from morphological and molecular analyses (18S rDNA and 28S rDNA sequences) of single-picked nematode specimens in accordance with current research standards. The aim of the present study is to evaluate nematode diversity and their phylogenetic relationships in a little-studied area, in consideration of their importance for potential downstream use as bioindicators of coral reefs. The results will help establish a baseline for nematode diversity research for coral reefs around Okinawa-jima Island, and help generating new sequences for genera not registered in the common databases.

Materials & Methods

Sites and sampling strategy

Sediment samples were collected in the infralittoral area of two sites around Okinawa-jima Island (Japan, Figs. 1A, 1B.); Senaha Beach (26.4247°N, 127.7342°E, Figs. 1C, 1D) and Kouri-jima (26.7067°N, 128.0181°E, Figs. 1E, 1F). Samples were collected during low tide at depths between 0.8 m and 2.1 m (Table 1) between October 15 to November 11, 2022. These two locations were selected because their micro-habitat complexity over a small area was ideal to make comparisons within each site; with seagrass, bare sand, and coral outcrops all within 800 m of each other. Latitude and longitude coordinates were taken by using a handheld GPS device (Meiri, 2018; Table 1). Figure 1 was created using QGIS (Version 3.36.2, QGIS Development Team, 2024), using the QuickMapServices plugin with Google Maps (https://maps.google.com) satellite layer.

Figure 1 Map of Okinawa-jima Island with sampling sites in this study.

Map of Japan and Okinawa-jima Island (A, B) Transects at Senaha Coast (C, D) and at Kouri-jima Island (E, F). The images were obtained through open source data on QGIS.

At each microhabitat within both sites, three sediment samples were collected every 10 m each from sea grass-dominated areas, bare sand areas, and coral outcrops (Figs. 1C, 1D, 1E; Table 1), making a total of 9 collected sediment samples per site and 18 overall. Sediment samples were used for morphological and molecular analyses and were obtained by pushing the manual corer tube (four cm Ø, 10 cm H) vertically into the sediment. Sediments were then fixed into a mixture of 4% formalin and rose Bengal or 80% ethanol, and stored at −4 °C.

All sediment samples were washed and processed using colloidal silica Ludox® (DuPont) HS 40 as a high-density solution (Burgess, 2001). The sediment was then filtered through a two mm mesh sieve to eliminate coral rubbles and rocks. Each sample was divided and transferred into four Falcon® tubes that were filled with a mix of two parts of Ludox® and three parts of distilled water, centrifuged at 3,000 g for 10 min and washed again. Finally, the supernatant was filtered with a 45 µm sieve. This process was repeated three times to maximize the number of nematode specimens collected.

Table 1 Coordinates of sampling locations around Okinawa-jima Island in this study.

Site	Habitat	Replicate code	Latitude	Longitude	Depth (m)	Date	Observations	
Senaha	Sand 1	NS1	26.42474	127.73393	1.0	10/21/2022	 	
Senaha	Coral 1	NC1	26.42667	127.73485	1.8	10/22/2022	Coral bleaching observed	
Senaha	Seagrass 1	NF1	26.42483	127.73396	1.0	10/21/2022	 	
Senaha	Sand 2	NS2	26.42477	127.73385	1.0	10/21/2022	 	
Senaha	Coral 2	NC2	26.42669	127.73493	2.1	10/22/2022	Coral bleaching observed	
Senaha	Seagrass 2	NF2	26.42487	127.73387	1.0	10/21/2022	 	
Senaha	Sand 3	NS3	26.42492	127.73378	1.0	10/21/2022	 	
Senaha	Coral 3	NC3	26.42678	127.73490	2.0	10/22/2022	Acropora spp. present	
Senaha	Seagrass 3	NF3	26.42493	127.73380	1,0	10/21/2022	 	
Kouri	Sand 1	KS1	26.69428	128.02122	1.1	15/10/2022	 	
Kouri	Coral 1	KC1	26.69958	128.02738	0.8	08/11/2022	Acropora spp. present	
Kouri	Seagrass 1	KF1	26.69437	128.02314	1.5	15/10/2022	 	
Kouri	Sand 2	KS2	26.69424	128.02113	1.0	15/10/2022	 	
Kouri	Coral 2	KC2	26.69965	128.02750	1.0	08/11/2022	Acropora spp. present	
Kouri	Seagrass 2	KF2	26.69441	128.02322	1.4	15/10/2022	 	
Kouri	Sand 3	KS3	26.69419	128.02105	1.0	15/10/2022	 	
Kouri	Coral 3	KC3	26.69971	128.02758	1.0	08/11/2022	Acropora spp. present	
Kouri	Seagrass 3	KF3	26.69446	128.02329	1.3	15/10/2022	 	

Environmental data collection

To reduce potential biases due to environmental factors at the sites, environmental variables and anthropic pressures and impacts were measured. Additional sediment and seawater samples were collected from each habitat. Seawater was taken using a simple bucket (24 × 24 cm) from the surface in each location and the sediment was collected with cylindrical manual corers (4 cm Ø, 15 cm H) at the seafloor (Table 1). We analyzed water quality parameters using a RINKO probe (data for temperature, salinity, conductivity, dissolved oxygen, turbidity, EC25, density, chlorophyll fluorescence, chlorophyll-a; Table 2). From the collected sediment, we conducted a particle size analysis (PSA) on each sample using a granulometric analyser (Particle Image Analyzer; JASCO Labs FF-30Micro) to determine granulometry ISO mean and the percentages of silt/clay (ca < 63 µm), sand (ca 63–2,000 µm), and gravel (ca > 2,000 µm), as defined by Blott & Pye (2012). These analyses were performed one time per replicate, and thus three times for each microenvironment for each site (total n = 18).

Table 2 Means of the environmental variables in each site and habitat.

 	 	Senaha_seagrass	Senaha_sand	Senaha_coral	Kouri_seagrass	Kouri_sand	Kouri_coral	
In-situ evaluation	Date of sampling	21/10/2022	21/10/2022	22/10/2022	15/10/2022	15/10/2022	08/11/2022	
 	Locality	Senaha Beach	Senaha Beach	Senaha Beach	Kouri-jima	Kouri-jima	Kouri-jima	
 	Tide at sampling	Low	Low	Low	Low	Low	Low	
Granulometry machine measures	ISO inner diameter mean (µm)	294.82	450.75	2,263.63	1,466.67	767.50	1,168.07	
 	Silt/clay	0.37	0.00	0.00	0.00	0.00	0.00	
 	Sand	98.37%	97.76%	73.50%	82.37%	92.80%	92.80%	
 	Gravel	1.26%	8.90%	25.38%	17.63%	7.20%	7.20%	
RINKO probe	Temp (deg C)	28.43	28.43	29.08	28.88	28.96	28.19	
data	Sal (PSU)	32.69	32.69	33.40	28.90	20.08	24.16	
 	Cond (mS/cm)	53.35	53.35	55.24	48.20	34.77	40.49	
 	EC25 (uS/cm)	49,600.86	49,600.86	50,692.26	44,406.31	31,978.81	37,836.11	
 	Density (kg/m3)	1,020.52	1,020.52	1,020.94	1,017.53	1,010.91	1,014.21	
 	Chl-Flu, (ppb)	0.77	0.77	0.46	0.34	0.43	0.34	
 	Chl-a (ug/l)	0.77	0.77	0.46	0.34	0.43	0.34	
 	Turb-M (FTU)	2.63	2.63	1.70	1.13	2.08	1.38	
 	DO (%)	132.03	132.03	118.60	101.91	110.82	102,16	
 	Weiss-DO (mg/l)	8.52	8,52	7.56	6,564.79	7,596.66	6,944.17	

Additionally, we used the “DiBattista scale” (DiBattista et al., 2020) to estimate the level of cumulative anthropic pressure and impact present for both sites investigated. DiBattista et al. (2020) analyzed the impact of natural and anthropogenic pressures on selected sampling sites in Okinawa-jima Island via a point-based assessment system in order to rank sites based on cumulative anthropogenic impacts. Criteria considered in this scale included distance from shore, distance from heavily populated areas, freshwater input, fishing pressure, coastal development, presence of human recreation area, hermatypic coral cover, and other notable pressures. Scored sites were then grouped into low (score = 10 to 7), medium (score = 7 to 4), or high (score = 4 to 0) anthropogenic pressure groups.

Morphological analyses

A total of 100 nematodes (or all available individuals if fewer than 100) were randomly selected from each replicate using a stereomicroscope (LEICA S8AP0). The nematodes obtained were mounted on permanent slides by using a simplified version of the Seinhorst (1962) method and identified with an optical microscope (Nikon ECLIPSE 80i), using a maximum magnification of 100X. For identification, we used the pictorial keys from Platt & Warwick (1983), Platt & Warwick (1988), Warwick, Platt & Somerfield (1998), Wieser (1953), the World Database of Nematodes Nemys (Nemys, 2024; https://nemys.ugent.be/), and papers including the most up-to-date information relating to the taxa found (Ahmed, Boström & Holovachov, 2020; Fadeeva, Mordukhovich & Zograf, 2016; Kotta & Boucher, 2001; Leduc, 2013; Leduc & Sinniger, 2018; Leduc & Zhao, 2021; Muthumbi & Vincx, 1999).

The data obtained through morphological analyses were examined using the software R (R Core Team, 2025). The Chao 1 (Chao, 1984) and Shannon (Shannon, 1948) indices were calculated and confronted (Oksanen et al., 2025; Wickham, 2016; Wickham, 2025)

Additionally, a multivariate analysis of variance based on permutations (PERMANOVA; Anderson, 2001) was performed on transformed data to calculate the significant variance in nematode assemblages among sites and habitats. A similarity of percentage (SIMPER) analysis was used to determine which taxa influenced the spatial distribution, and a PERMDISP analysis was also conducted to test the null hypothesis among the analyzed environmental parameters. These analyses were performed with PRIMER-e 7 + PERMANOVA (Anderson, 2001).

The environmental data (depth, DiBattista scale information, ISO inner diameter mean, silt/clay%, sand%, gravel%, as well as temperature, salinity, conductibility, chlorophyll-a and dissolved organic matter of seawater) were imported and transformed using the function “(x-mean)/stdev”. To visualize possible differences between the structure of the communities and to clarify the relationship between environmental data and abundance, a Distance-based multivariate multiple regression (DistLM), a principal component analysis (PCA), and a non-metric multidimensional scaling (nMDS) analysis were performed on the selected database. The nMDS analysis was performed using Bray-Curtis dissimilarity and log-transformation to reduce the influence of abundant taxa. Statistical analyses and graphics were obtained using the software Past 4.03 (Paleontological Statistics Software Package for Education and Data Analysis; Hammer, Harper & Ryan, 2001).

DNA extraction, PCR conditions, and sequencing

Nematode specimens were randomly picked from each replicate with the same method used for the morphological analyses. The organisms were individually picked using a sterile syringe, placed on a microscopy slide, rapidly identified and finally transferred singularly into 1.5 ml tube. The slides were cleaned between every specimen using bleach and heat sterilization by flaming and rinsed with ethanol to remove any residues from the surface. We used the commercial kit Qiagen DNeasy Blood and Tissue Kit (QIAGEN, Hilden, Germany) following the manufacturer’s standard protocol (Haendiges et al., 2020) for DNA extraction. To avoid an initial low percentage of successful extractions, we repeated the final elution step in the AE buffer two times for a total volume of 50 µl of DNA.

To assess trends and standards in free-living marine nematode phylogenetic research, we searched literature using Google Scholar and ISI Web of Science in January 2024. We used the key words; “free-living”, “marine”, “nematode” and “phylogeny” or “phylogenetic trees” or “evolution” and excluded “freshwater”, “soil”, “terrestrial” and “parasite” to avoid including research conducted only on parasitic nematodes or free-living terrestrial or freshwater nematodes. We found seven peer-reviewed publications matching our criteria and summarized them to highlight markers and primers commonly used in phylogenetic analyses until now (Supplemental Information 8; Armenteros et al., 2014; Avó et al., 2017; Bik et al., 2010a; Bik et al., 2010b; Derycke et al., 2010; Kumar, Sen & Bhadury, 2014; Pereira et al., 2010). Based on these results, for this study we chose to amplify 18S rDNA and 28S rDNA, as they have been proven to be effective for a wide range of different taxa within the phylum Nematoda.

Target gene fragments were then amplified via PCR. Each 25 µl PCR reaction for the two markers consisted of 2.5 µl of 10X Standard Taq Reaction Buffer, 0.5 µl of 10 mM dNTPs, 0.5 µl of forward primer, 0.5 µl of reverse primer, 0.125 µl of Taq DNA polymerase, 1.0 µl of MgCl, 18.375 µl of ultra-pure RNA-free water and 1.5 µl of raw DNA.

To amplify 18S rDNA, we used the primer pair Nem18SF (5′-ATTCCGATAACGARCGA GAC-3′)/Nem18SR (5′-CCGCTKRTCCCTCTAAGAAGT-3) (Wood et al., 2013) and for 28S rDNA, D2A (5′ ACAAGTACCGTGAGGGAAAGT3′) and D3B (5′ TGCGAAGGAACCAGCTACTA 3′) (Nunn, 1992; Zhao et al., 2008). The optimized PCR conditions were an initial denaturation of 5 min at 94 °C, followed by 40 cycles of 94 °C for 30 s, 54 °C for 30 s, 72 °C for 1 min, then followed by a final extension of 10 min at 72 °C. PCR products were visualized on a 1.2% agarose gel, and successful amplifications were purified using exonuclease and shrimp alkaline phosphatase digestion, and sent for Sanger sequencing in both directions to FASMAC (Kanagawa, Japan).

Phylogenetic analyses

New sequences obtained were edited in Geneious (Geneious Prime 2023.2; Kearse et al., 2012) using chromatograms to assess their quality. Short sequences (<500 bp) and/or those of low sequence quality (<80%) were discarded. Forward and reverse sequences were assembled and trimmed with the same software. We used BLAST (Basic Local Alignment Tool; Altschul et al., 1990) to search individually for nucleotide sequence homology and MEGA11 (Molecular Evolutionary Genetics Analysis version 11; Tamura, Stecher & Kumar, 2021) to create a reference database of the taxa identified. In addition, previously reported 18S rDNA (n = 66) and 28S rDNA (n = 16) sequences of marine nematodes from GenBank were included in our respective databases (Supplemental Information 1). All sequences were aligned using MUSCLE (MUltiple Sequence Comparison by Log-Expectation; Edgar, 2004) as implemented in MEGA11 with default parameters and the ends of the resulting alignments were trimmed.

DnaSP v6.12.03 (Rozas et al., 2017) was used to obtain a list of the potential haplotypes among all sequences amplified (Supplemental Information 2), which were uploaded on MEGA11 (Tamura, Stecher & Kumar, 2021) to select the best nucleotide substitution model (GTR+G for the 18S sequences and GTR+G+I for the 28S sequences), align the sequences, and create a maximum likelihood tree (bootstrap substitution rate = 1,000).

In addition, gene trees were reconstructed with the same dataset by applying Bayesian Inference (BI) on MrBayes version 3.2.7 (Ronquist et al., 2012), through the CIPRES Science Gateway (Miller, Pfeiffer & Schwartz, 2010).

The BI trees were re-rooted in order to separate the two classes of the phylum Nematoda (Chromadorea and Enoplea) and the outgroup pruned to have a clearer view of clades (see original trees in Supplemental Information 3–6). The obtained trees were visualized with FigTree version 1.4.4 (Rambaut, 2018) and edited with Inkscape version 1.3.2 (Inkscape Project, 2020, https://inkscape.org/).

Comparison of methods

We used a Venn diagram, elaborated with R (R Core Team, 2025) and using indications from Gao (2023) and Yan (2023), to visualize the similarity and efficiency among our classic morphological identification and DNA barcoding of the 18S rDNA and 28S rDNA markers.

Results

Granulometry and environmental variables

All sediment samples (n = 6) were characterized by the dominance of sand (73.50%–98.37%), followed by gravel and clay fractions, with a mean grain diameter ranging from 294.82 µm in Senaha_seagrass to 2,263.63 µm in Senaha_coral (Table 2). On average, seawater temperatures were stable, ranging from 28.19 °C to 29.08 °C. Salinity, conductivity and chlorophyll-a level were higher in Senaha habitats than at Kouri-jima (Table 2).

Morphological results

Focusing on the phylum Nematoda, 24 families were found, with the top five families in terms of abundance (Chromadoridae, Desmodoridae, Cyatholaimidae, Linhomoeidae and Oncholaimidae) contributing more than 60% of the total nematode numbers (Fig. 2). The total dataset of all samples consisted of 104 nematode taxa, of which 80 were identified to the level of genus, one to subfamily and 23 to family (Supplemental Information 7). The three most common taxa in the class Enoplida were Pareurystomina sp. (4.38%, n = 60), Viscosia sp. (3.58%, n = 49), and Enchelidiidae indet. (2.55%, n = 35), while in the class Chromadorea, we found Linhomoeidae indet. (5.99%, n = 82), Vasostoma sp. (4.38%, n = 60) and Daptonema sp. 2 (3.94%, n = 54). Daptonema sp. 2, Pareurystomina sp. and Enchelidiidae were the most common taxa in Senaha_sand; Daptonema sp.1, Pareurystomina sp. and Viscosia sp. in Senaha_coral; Linhomoeidae, Paracanthonchus sp. and Didelta sp. in Senaha_seagrass; Vasostoma sp., Linhomoeidae and Prochromadorella sp. in Kouri_sand, Epsilonema sp. Monoposthiidae sp. and Eurystomina sp. in Kouri_coral and Linhomoeidae, Molgolaimus sp. and Parapinnanema sp. in Kouri_seagrass.

Figure 2 Abundance percentage nematode families.

N, Senaha; K, Kouri; M, morphology; F, seagrass; S, sand; C, coral. Numbers refer to replicate.

The scatter plot of Chao1 and the Shannon indices are represented in Fig. 3. The Chao1 values in our samples ranged from 8 (KMC1) and 41 (NMF2), indication of an overall substantial difference in terms of species richness between samples. NMF2 (Chao1 = 41), NMS1 (Chao1 = 36) and KMF1 (Chao1 = 33) were the richest samples, as they included more rare taxa, while KMC1 (Chao1 = 8), KMC2 (Chao1 = 13), KMC3 (Chao1 = 17) were the least rich. The Shannon values, instead, ranged from 1.93 (KMS1) to 3.43 (NMF2), demonstrating a significant variation in both richness and evenness. The most diverse samples were NMF2 (Shannon = 3.43), NMS1 (Shannon = 3.30) and NMF1 (Shannon = 3.20), and the least diverse KMS1 (Shannon = 1.93), KMC1 (Shannon = 2.02) and KMC3 (Shannon = 2.04), due to the dominance of the genus Epsilonema. The relationship between Chao1 and Shannon was usually proportional, with the exceptions of NMC2 (Chao1 = 31 and Shannon = 2.79) and KMF1 (Chao1 = 33 and Shannon = 3.15), which had similar richness but different Shannon values, indicating that they tended to be dominated by less taxa.

Figure 3 Chao1 Index and Shannon Index scatter plot.

N, Senaha; K, Kouri; M, morphology; F, seagrass; S, sand; C, coral. Numbers refer to replicate.

The PERMANOVA main-test was significant for the factors Habitat, Site and Habitat × Site (for all, pPerm < 0.01). However, the PERMDISP analysis was not significant, indicating that the results of the PERMANOVA were not caused by heterogenic multivariate dispersion.

SIMPER analysis showed that the average similarity in group N (Senaha) was higher (52.06%) than in group K (Kouri; 41.80%), and that the main taxa responsible for this difference were Spirophorella sp., Viscosia sp. and Croconema sp. in group N, and Linhomoeidae, Daptonema sp. 1 and Epsilonema sp. in group K. The average dissimilarity between the two groups was 79.93% due to the higher abundance of Viscosia sp. and Linhomoeidae.

Focusing on habitats, the average similarity in coral samples was lower (41%) than in sand or seagrass, due to the presence of Daptonema sp. 1, Spilophorella sp. and especially Epsilonema sp., which was found almost only in Kouri_coral. Finally, in terms of dissimilarity, coral and seagrass showed the highest dissimilarity (81.58%), followed by sand and coral (70.82%), and finally sand and seagrass (64.99%). The main taxa determining the dissimilarity overall were Vasostoma sp., Daptonema sp. 1, Daptonema sp. 2, Chromadoridae, Enchelidiidae and Molgolaimus sp.

The PCA (Fig. 4A) highlighted a clear separation between microhabitats mainly based on two parameters: depth and DiBattista scale. To assess the influence of these environmental variables on species composition, we also performed a DistLM analysis (Fig. 4C) marginal test (R2, specified), which reveals that the environmental variables of the DiBattista scale, ISO inner diameter mean, silt/clay%, temperature, salinity, conductibility, chlorophyll-a and dissolved organic were significant (P < 0.05). Another DistLM analysis (AICc, step-wise) also highlighted the importance of the DiBattista scale and depth as the two main factors determining the composition of nematode assemblages. The nMDS ordination (Fig. 4B) plot was elaborated including the significant environmental factors found by the DistLM. The final stress value of 0.1034 indicated the plot provided a good graphic representation of the data. The resulting plot (Fig. 4D) revealed a significantly divergent clustering of the samples in accordance with the habitat type, and coral reefs clustering separately from the others. PCA and nMDS showed similar orientations, especially for the coral reef samples from both sites.

Figure 4 Statistical analyses based on morphological results.

(A) Principal component analysis (PCA) showing the relationship between environmental variables and sites. (B) Non-metric multidimensional scaling (nMDS) showing the relationship between environmental variables, diversity and sites. (C) Distance-based multivariate multiple regression (DistLM) showing the two main parameters responsible for the variation. (D) Hieratical classic clustering of the replicates showing how samples Kouri_coral were clearly separated from the others.

Molecular results and phylogeny

18S rDNA

The 18S rDNA alignment consisted of 243 sequences of 767 base pairs, including 66 nematode and 2 non-nematode sequences that were retrieved from NCBI as references (S1), and 175 newly obtained sequences from this research. After selecting haplotypes with DNASP6 (S2), we obtained 104 haplotype sequences for the marker 18S which were deposited in GenBank under accession numbers: PP859916–PP859943, PP859945–PP859974, PP859976–PP859979, PP859981–PP859994, PP859996–PP860023 (S2). Among these, five sequences corresponded to taxa non previously reported in NCBI (1 x Anonchus Cobb, 1913; 1 x Filoncholaimus Filipjev, 1927; 1 x Nannolaimoides Ott, 1972; 2 x Parapinnanema Inglis, 1969 (accession numbers: PP859916, PP860023, PP859969, PP859948 and PP859949 respectively). The sequences used for the final alignment correspond to 49 taxa (1 suborder, 11 families, and 37 genera): Acanthopharynx (3), Anonchus (1), Anticoma (2), Aponema (1), Axonolaimidae (1), Axonolaimus (2), Catanema (2), Cephalanticoma (3), Cheironchus (1), Chromadorella (2), Chromadoridae (1), Cobbia (1), Cyatholaimus (3), Daptonema (2), Desmodora (30), Desmodoridae (2), Desmolaimus (1), Enchelidiidae (1), Enoploides (1), Epacanthion (3), Eurystomina (1), Filoncholaimus (1), Fotolaimus (1), Laxus (3), Linhomoeidae (1), Mesacanthoides (1), Meyersia (34), Microlaimidae. (1), Microlaimus (2), Monhysterina (1), Nannolaimoides (1), Neochromadora (1), Oncholaimidae (3), Paracanthonchus (11), Parapinnanema (2), Pareurystomina (1), Phanodermatidae (4), Rhabdocoma (1), Rhabdodemania (1), Selachinematidae (4), Spirinia (1), Steineria (1), Theristus (1), Thoracostomopsidae (3), Trischistoma (1), Vasostoma (3), Viscosia (2), Xyalidae (5), Zalonema (7).

The inferred phylogenetic trees exhibited similar topologies for both BI and ML analyses (Supplemental Information 3 and 4), with a clear, although not strongly supported, division between classes Chromadorea and Enoplea (Fig. 5). In total, 19 families (highlighted by different colors in Fig. 5) were recovered by the phylogenetic tree for the 18S marker generated by Bayesian Inference, of which nine were within the class Chromadorea (Xyalidae, Comesomatidae, Axonolaimidae, Linhomoeidae, Selachinematidae, Desmodoridae, Chormadoridae, Microlaimidae and Cyatholaimidae), and 10 within Enoplea (Aphanolaimidae, Rhabdodemaniidae, Trischistomatidae, Trefusiidae, Oncholaimidae, Enchelidiidae, Anticomidae, Enoplidae, Thoracostomopsidae and Phanodermatidae). Within the class Chromadorea, the clade formed by the orders Araeoloimidae+Monhysterida was supported as a separate group, similarly for the clade Chromadorida+Desmodorida (to the exception of Cheironchus sp., which was placed between the families Linhomoeidae and Desmodoridae). All the Chromadorea families were supported by at least one of the two analytical methods with the exception of Selachinematidae, which was split into two in both BI and ML analyses (Fig. 5, and Supplemental Information 3 and 4).

Figure 5 Phylogenetic tree for the 18S marker, generated by Bayesian inference.

Numbers on the branches represent posterior probability and bootstrap values. below 0.97 and 75% respectively are not shown. (A) Selective deposit-feeders; (B) non-selective deposit-feeders; (A) epistrate-feeders; (B) omnivores/predators (Wieser, 1953).

In the class Enoplea, only families Anticomidae and Enoplidae were recovered as well supported groups (posterior bootstrap, b = 0.97; probability, pp = 96% respectively; Fig. 5), while families Oncholaimidae and Enchelidiidae were placed into the same group. The families Oncholaimidae and Enchelidiidae appeared to be in the same clade and they were supported by both BI (value = 1) and ML (percentage = 100%).

28S rDNA

The 28S rDNA alignment consisted of 82 sequences of 720 base pairs, including 18 sequences that were retrieved from NCBI as references (Supplemental Information 1) and 64 new nematode sequences. After selecting haplotypes with DNASP6 (Supplemental Information 2), we obtained 37 sequences, none of which had been previously reported in NCBI (accession numbers: PP859885–PP859915). Of these, 32 were selected by the Bayesian Inference (BI) analysis to generate the 28S final tree. The number of sequences for each taxon corresponds to the following: Anticoma (1), Anonchus (1), Axonolaimidae (1), Cephalanticoma (1), Chromadoridae (1), Chromadorita (5), Cobbia (1), Daptonema (2), Demonema (1), Desmodora (11), Desmodoridae (3), Desmodoroidea (1), Desmolaimus (1), Enoploides (1), Epacanthion (2), Eurystomina (1), Meyersia (14), Monhysterina (1), Oncholaimidae (1), Paracanthonchus (5), Phanodermatidae (5), Rhabdodemania (1), Viscosia (1), Zalonema (2), which includes 1 suborder, 1 superfamily, 5 families, and 17 genera.

The same methods (BI and ML) were used to reconstruct the trees for the 28S marker (Fig. 6, Supplemental Information 5 and 6), and the topologies also appeared to be similar in both analytical approaches. For 28S rDNA, eight families were found as part of the class Chromadorea (Desmodoridae, Chromadoridae, Cyatholaimidae, Selachinematidae, Xyalidae, Axonolaimidae, Comesomatidae and Linhomoeidae, but not Microlaimidae) and six in the class Enoplea (Thoracostomopsidae, Phanodermatidae, Aphanolaimidae, Rhabdodemaniidae, Enchelidiidae and Oncholaimidae, but not Trischistomatidae, Trefusiidae, Anticomidae or Enoplidae). Only the families Xyalidae, Enchelidiidae and Oncolaimidae formed well-supported clades by both BI (pp = 1) and ML (b =99%).

Figure 6 Phylogenetic tree for the 28S marker, generated by Bayesian Inference.

Numbers on the branches represent posterior probability and bootstrap values. Values below 0.97 and 75% respectively are not shown. 1A, selective deposit-feeders; 1B, non-selective deposit-feeders; 2A, epistrate-feeders; 2B, omnivores/predators (Wieser, 1953).

Comparison of methods

The results obtained from the three methods (morphology, 18S rDNA barcoding, 28S rDNA barcoding) were compared with a Venn diagram to show the overlap in taxa identification between the different methods (Fig. 7).

Figure 7 Venn diagram showing the results of the identification obtained with the three methods (morphology, DNA barcoding 18S, DNA barcoding 28S).

54 operational taxonomic units (OTUs) were identified only by morphology, including three genera determined from the SIMPER analysis [Spilophorella sp. (Chromadoridae), Molgolaimus sp. (Desmodoridae), and Epsilonema sp. (Epsilonematidae)]. On the other hand, 12 OTUs were amplified only by the 18S rDNA marker, two only by the 28S rDNA marker, 25 by morphology + 18S rDNA, two by morphology + 28S rDNA and one by 18S rDNA + 28S rDNA. Among the diversity of these, Acanthopharynx sp. (Desmodoridae), Laxus sp. (Desmodoridae), and the family Thoracostomopsidae was underestimated by morphological methods. Only 21 OTUs were identified or amplified by the three methods overall, and most of the corresponding sequences of both markers did not have high levels of similarity (e.g., >95%) to sequences already deposited in GenBank.

Discussion

This study aimed to obtain a working framework of the diversity of the phylum Nematoda in two coastal coral reef sites on the west coast of Okinawa-jima Island (Senaha coast and Kouri-jima) across three habitats (sand, coral reef, seagrass beds) in order to investigate the spatial distribution of nematodes. We used different methods, a “classic” morphological taxonomic approach, and DNA barcoding of two markers, and compared the results. Our resulting dataset showed that the shallow water nematode assemblages recovered from intertidal coral reef areas of Okinawa-jima Island comprise a taxonomically diverse range of genera and families.

Integration of morphology and DNA barcoding

The taxonomy of nematodes has always relied on morphology (Nisa, Tantray & Shah, 2022). Our molecular analyses provided partial support for the morphological identification of the free-living nematodes in this study. Overall, all methods recovered the same main families (Chromadoridae, Desmodoridae, Cyatholaimidae, Linhomoeidae and Oncholaimidae). However, discrepancies arose with the less common taxa, and morphological analyses revealed greater diversity than molecular results. Similar results have been obtained in other studies that reported higher diversity through morphology rather than molecular methods, including Abad et al. (2016) from plankton specimens of estuarine water, Dell’Anno et al. (2015) in a deep-sea study, Harvey et al. (2017) from zooplankton samples, Schenk, Kleinbölting & Traunspurger (2020) from nematodes, and Stefanni et al. (2018) from copepods. We hypothesize that low amplification success in some specimens may be due to the prevalence of small individuals, such as juveniles, or due to the small sizes of individuals of families like Desmoscolecidae and Epsilonematidae. Additionally, high nucleotide variability and indels at primer sites (Creer et al., 2010), a documented issue in free-living nematodes (Bhadury et al., 2006; De Ley et al., 2005), could have contributed to amplification failure in certain groups. The discrepancies between methods may also arise, at least in part, from the overestimation of congeneric specimens caused by the morphological plasticity typical of families such as Xyalidae and Chromadoridae. Conversely, some families, including Desmodoridae and Thoracostomopsidae, exhibit meristic features that are difficult to distinguish morphologically. For these groups, molecular methods, particularly 18S rDNA analysis, proved to be more effective. Finally, discrepancies between our morphological identifications and the closest matches in available molecular databases are most likely the result of the limited taxonomic coverage of the sequences available in GenBank relative to the vast diversity of marine nematodes (Lambshead & Boucher, 2003; Pereira et al., 2010), and the lack of a reference database from Okinawa-jima Island. Considering the sometimes-insufficient number of reference species and genera sequences deposited in GenBank (Charrier et al., 2024; Holovachov, Camp & Nadler, 2015), amplifying the available dataset of 18S and 28S genes (as they are widely used markers for invertebrates phylogenetics; (Holovachov, Camp & Nadler, 2015; Mallatt, Garey & Shultz, 2004) is essential to improve the understanding of nematode phylogeny and evolutionary history (Charrier et al., 2024). A more complete dataset (Charrier et al., 2024; Holovachov, Camp & Nadler, 2015) would improve the taxonomic coverage of underrepresented taxa or regions, reducing, in some cases, misidentifications and identification at high taxa, and speeding up the process of describing and integrating novel species in the phylogenetic tree by reducing stochastic errors (Holovachov, Camp & Nadler, 2015; Mallatt, Garey & Shultz, 2004) and making alignments more robust and better supported (Holovachov, Camp & Nadler, 2015; Mallatt, Garey & Shultz, 2004; Subbotin et al., 2008).

Diversity and microhabitats

There is a strong correlation between environmental conditions and the composition of nematode assemblages in coral reef areas (Ruiz-Abierno & Armenteros, 2016). Among the most relevant local drivers granulometry (Gourbault & Renaud-Mornant, 1990; Heip, Vincx & Vranken, 1985; Kotta & Boucher, 2001; Netto, Warwick & Attrill, 1999; Semprucci et al., 2011), pH levels (Esteves et al., 2022), hydrodynamic conditions (Semprucci et al., 2018), periodic oxygen depletion (Boucher & Gourbault, 1990; Gourbault & Renaud-Mornant, 1990; Kotta & Boucher, 2001; Sournia, 1976), depth (Netto, Warwick & Attrill, 1999), and the presence of coral fragments (Netto, Warwick & Attrill, 1999; Raes, Decraemer & Vanreusel, 2008) have been mentioned in the literature.

In our study, we observed a positive correlation between the Shannon (which represents richness and evenness) and Chao1 (which represents richness) indices. This suggests that communities with a higher number of taxa also tended to exhibit greater evenness (Konopiński, 2020; Morris et al., 2014; Voutilainen & Kangasniemi, 2015). Samples from Senaha Beach generally had both higher Chao1 and Shannon compared to Kouri, except for KMF1 (Kouri_seagrass 1). While the seagrass habitats had higher diversity and richness, followed by Senaha_coral, sand habitats and Kouri_coral, our morphological results showed that coral reef habitats were different from sand and seagrass beds in terms of nematode generic assemblages. Coral reef habitats stood out in terms of dissimilarity from both sand and seagrass beds, and in particular Kouri_coral, while having low diversity compared to the other microhabitats, although they had unique nematode assemblages mainly due to the presence of Epsilonema, Spilophorella, Enchelidiidae, Daptonema, Molgolaimus, Viscosia, Linhomoeidae, and Vasostoma). In addition, PCA analyses and nMDS ordination clearly showed coral sites as different from the other sites in terms of environmental parameters and diversity, with the main two factors being depth (an artifact of our sampling design) and anthropogenic pressures/impacts. Even considering that this research was limited to only two sites, both coral habitats showed the lowest similarity among themselves and the highest dissimilarity from the other habitats. These results suggest that microhabitats can be a potential driver in structuring nematode diversity at least in our examined sites.

Morphological evidence from other parts of the world (Armenteros et al., 2009; Armenteros & Ruiz-Abierno, 2015; Semprucci et al., 2013; Semprucci et al., 2018) has revealed that the main genera appear to be ubiquitous worldwide. For example, the genus Epsilonema was also found in coral reef fragments from the Porcupine Seabight (NE Atlantic) by Raes & Vanreusel (2006) and it was one of three most common genera found in Kenya and Zanzibar by Raes et al. (2007). Spilophorella and Viscosia were two of the top three genera discovered in one site of a study conducted in Taiwan by Wei-Ling et al. (2022), and Spilophorella is described as having the potential to be used as an indicator for a healthy environment (Wei-Ling et al., 2022). Spilophorella was also recorded as a common genus, for example, in the central Great Barrier Reef (Tietjen, 1991) and in the Caribbean (Ruiz-Abierno & Armenteros, 2016) and in Shimoni (Kenya) by Hashim et al. (2022). The genus Daptonema is widely distributed in diverse substrates globally, including coral reefs (Wei-Ling et al., 2022; Grassi et al., 2022; Semprucci et al., 2013). Viscosia is another common and ubiquitous genus, and in a few sites was recorded as the only genus present (Mohammad, 2022; Samad et al., 2018) or as part of non-diverse communities (Sherman, 1985). Molgolaimus has been recorded from mangrove areas (e.g., Sun & Huang, 2024; Zhou et al., 2020), but not directly associated with coral reefs. Finally, there are no direct records of Vasostoma in coral reef areas, but the genus was found in three Indian harbors (Nanajkar & Ingole, 2010) and from the continental slope in New Zealand (Leduc & Nodder, 2012).

However, the most frequent taxa are generally ubiquitous, while numerous novel genera and species have been recently discovered in tropical areas including the East China Sea (Chen & Guo, 2014; Chen & Guo, 2015; Chunming, Liguo & Yong, 2015; Leduc, 2022; Leduc & Sinniger, 2018; Lu, Sui & Huang, 2022; Semprucci & Balsamo, 2014; Sun, Huang & Huang, 2018; Sun, Huang & Huang, 2021). Our results strongly suggest that it is worth investigating nematode communities in coral reefs much more closely in Okinawa-jima Island as they may be a possible hotspot of diversity.

Topology of the phylum Nematoda

Regarding the structure of the phylogenetic trees we recovered in this study, less OTUs were recovered by 28S rDNA when compared to 18S rDNA, but the two markers (18S rDNA, 28S rDNA) and both ML and Bayesian trees showed similar topologies (Figs. 4 and 5), in accordance with previous studies (Supplemental Information 8; Armenteros et al., 2014; Avó et al., 2017; Bik et al., 2010a; Bik et al., 2010b; Derycke et al., 2010; Kumar, Sen & Bhadury, 2014; Pereira et al., 2010). Our recovered general topology was coherent with those in earlier findings (Ahmed, Boström & Holovachov, 2020; Smythe, Holovachov & Kocot, 2019), although the orders Enoplea and Chromadorea, recovered in all analyses, were not well supported.

18S rDNA proved to be more efficient than 28S rDNA in obtaining good quality sequences and resolving nematode phylogeny at the family level. The 18S rDNA results highlighted the monophyly of the families Xyalidae, Comesomatidae, Axonolaimidae, Linhomoeidae and Chromadoridae. It should be noted that the monophyly of the family Desmodoridae is controversial, and in accordance with Armenteros et al. (2014), Kumar, Sen & Bhadury (2014) and Meldal et al. (2007), our ML analysis supported this group, while the Bayesian analysis did not, with Epsilonematidae and Draconematidae sequences placed in the same clade. Inferring relationships within the Desmodoroidea is limited by the often fragmentary and potentially erroneous nature of the available molecular data and this situation may be responsible for the unrealistic genetic distances observed within some species and genera (Leduc & Zhao, 2021). Within the family Desmodoridae, phylogenetic relationships among the genera were partly resolved. Our study supported the monophyly of the genera Acanthopharynx, Zalonema, Catanema and Laxus, in contradiction with Armenteros et al. (2014), but not the genus Desmodora, for which the phylogenetic position remained ambiguous, possibly due to misidentifications in past research. Based on the World Register of Marine Species (WoRMS Editorial Board, 2024), Desmodora is the most diverse genus of the family Desmodoridae, currently containing 70 valid species, and many changes have been made within the genus as the identification of members this genus is challenging due to a limited number of useful differential morphological characters (Mordukhovich et al., 2023).

In accordance with Bik et al. (2010b) and Pereira et al. (2010), our analyses also supported the hypothesis that Oncholaimidae and Enchelidiidae may not be two different families, as our 18S and 28S rDNA results placed them in the same clade. In addition, we observed that they were very similar morphologically, particularly regarding the shape of the mouth and their sizes.

Conclusions

As this is one of the first attempts to focus on nematode assemblages in the shallow marine waters of Okinawa-jima Island, we could not directly compare our overall diversity data with previous studies, but our results suggest that nematode diversity in Okinawa is overwhelmingly high it is currently difficult to accurately determine due to the lack of baseline data, which this research provides. We identified at least 10 orders and 38 genera-level groups of nematodes from our samples that only span two different sites, and it is highly likely these samples include undescribed taxa. As other research has suggested, we demonstrated that morphological and DNA barcoding methods tended to reveal somewhat different results, particularly for less prevalent nematode taxa, perhaps due to morphological misidentifications stemming from ontogenetic features and morphological plasticity, and also likely due to a lack of sequences deposited in GenBank. For this reason, an integrated approach is recommended when it comes to studying the diversity of nematodes in areas not yet known, such as Okinawa and the Ryukyus (Reimer et al., 2019).

For instance, the phylogenetic analysis revealed that most genera within the family Desmodoridae are monophyletic, while the genus Desmodora appears to be more complex and potentially polyphyletic. Finally, the structure of the trees we obtained offers insights into the topology of the phylum Nematoda, suggesting that Oncholaimidae and Enchelidiidae might not be two different families. Overall, our study showed Okinawan coral reefs and neighboring areas are potential hot-spots of diversity for free-living marine nematodes, and that nematode communities here consist of a taxonomically diverse group of genera and families. The data acquired from this study represents the start of a baseline reference database for nematodes in Okinawa-jima Island, and it is hoped that future studies will build on our current findings.

Supplemental Information

Supplemental Information 1 List of NCBI sequences used as reference to build the phylogenetic trees

Supplemental Information 2 List of haplotypes used to build the phylogenetic trees

Supplemental Information 3 Tree 18S_bayes only

Supplemental Information 4 Tree 18S ML only

Supplemental Information 5 Tree 28S bayes only

Supplemental Information 6 Tree 28S ML only

Supplemental Information 7 List of nematode taxa obtained through morphology

Supplemental Information 8 Table of papers related to molecular work on nematodes

We thank Prof. T. Ravasi for the loan of the multi-parametric RINKO probe, and Prof. Fujita Kazuhiko (UR) for sharing the Particle Image Analyzer (JASCO FF-30Micro). This research was conducted in the Molecular Invertebrate Systematics and Ecology Laboratory (MISE lab, UR) and the first author thanks all MISE members for their support. Comments from anonymous reviewers improved an earlier version of this manuscript.

Additional Information and Declarations

Competing Interests

Author Contributions

DNA Deposition

Data Availability

James Reimer is an Academic Editor for PeerJ. Part of the equipment used in this research was provided by Okinawa Institute of Science and Technology Graduate University and financial support was provided by OIST COI-Next funding.

Marilyn Carletti conceived and designed the experiments, performed the experiments, analyzed the data, prepared figures and/or tables, authored or reviewed drafts of the article, and approved the final draft.

Nuria Viñuela Rodríguez analyzed the data, prepared figures and/or tables, authored or reviewed drafts of the article, and approved the final draft.

Gaia Rossetti performed the experiments, authored or reviewed drafts of the article, and approved the final draft.

Virginia Rossi performed the experiments, authored or reviewed drafts of the article, and approved the final draft.

Bryan Gabriel Pulido Tan performed the experiments, authored or reviewed drafts of the article, and approved the final draft.

James Davis Reimer conceived and designed the experiments, prepared figures and/or tables, authored or reviewed drafts of the article, and approved the final draft.

The following information was supplied regarding the deposition of DNA sequences:

The newly obtained DNA sequences are available at GenBank: PP859885–PP859943, PP859945–PP859974, PP859976–PP859979, PP859981–PP859994, PP859996–PP860023.

The following information was supplied regarding data availability:

The raw data are available in the Supplementary Files.

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
