# Peer review of "The tip of the iceberg: extraordinarily high diversity while examining two infralittoral nematode communities on Okinawa-jima Island, Japan, using morphology and DNA barcoding"

_PeerJ, doi:10.7717/peerj.19757_

## Round 0.1 · original submission · Minor Revisions

Please, take into account all the reviewers' comments and corrections. Note that there are some suggestions for the methodological part of the study.

Reviewer 1 ·

Basic reporting

This study contributes new DNA sequences to molecular databases that require further expansion, which could enhance research on the global diversity of free-living marine nematodes.

Overall, the manuscript is well-structured; however, I recommend excluding information on meiofaunal groups, as the study focuses on marine nematodes. Including other taxonomic groups may confuse.

Additionally, I suggest improving some paragraphs (highlighted in the document) to enhance readability and flow.

In the results section, it is necessary to clarify the number of specimens sequenced per genus.

Furthermore, the accession numbers correspond to 99 sequences, but the text mentions 149—this discrepancy should be clarified.

Tables 1 and 3 contain similar information; I suggest consolidating all relevant data into Table 3.

Table 2 appears unnecessary, as the background information has already been cited. If the authors wish to include it, it can be placed in supplementary material.

For Figure 1, the satellite images are appropriate; however, I recommend adding a map to indicate the global location of Okinawa-jima Island, Japan.

For Figure 2, I suggest removing the meiofauna histogram and replacing it with a histogram showing the identified nematode genera/species.

Regarding Figure 5, the branch names include accession numbers, which I assume correspond to NCBI sequences. However, it is unclear where the sequences generated by the authors are located in the phylogenetic tree—this should be clarified.

Lastly, the accession numbers listed in the manuscript are not yet accessible on NCBI.

Experimental design

To enhance the methods, provide detailed information on how the nematodes were handled during the molecular process to prevent contamination.

The data presented are statistically robust and demonstrate quality control in the genetic sequences obtained; however, some details highlighted in the document need clarification.

Considering these recommendations, I believe this research aligns well with the aims and scope of the journal.

Validity of the findings

The data reveal that coral reefs harbor the highest nematode diversity, with statistically significant differences from nearby sand and seagrass beds. On the other hand, the high dissimilarity observed suggests the presence of unique nematode assemblages in the coral reef. This information could improve our understanding of ecosystem dynamics, where marine nematodes play a crucial role. Particularly in this study will be interesting to know the implications of a higher presence of predators in the coral reef.
This study provides molecular evidence supporting the need for a future revision of the Oncholaimidae and Enchelidiidae families, which were classified here as a single family.

The results also contribute to the revision of the genus Desmodora, as the absence of distinct morphological characters in the genus suggests the presence of cryptic species that can be identified through molecular data.

Additionally, the findings could support re-evaluating the taxonomic position of Cheironchus sp., currently classified within the family Selachinematidae, but in this study, molecular analysis placed it between the families Linhomoeidae and Desmodoridae.

Additional comments

The paper contributes to information for future studies, It is recommended that the paper be resubmitted after the revision process.

Annotated reviews are not available for download in order to protect the identity of reviewers who chose to remain anonymous.

Reviewer 2 ·

Basic reporting

This study provides valuable insight into the nematode community in the coral reefs of Okinawa-Jima Island, utilizing both morphological and molecular techniques. The authors' findings highlight the high diversity of nematode species in the coral reef habitats.
There are several points that the authors could expand upon or clarify.
1. The comparison of different microhabitats reveals that the coral reef habitat supports the highest nematode diversity. While this is an interesting finding, it would benefit the authors to further elaborate on this observation by comparing it to similar studies in other coral reef ecosystems. Are the nematode species found in the Okinawa-Jima coral reef habitat and coral reefs from different regions? This would help establish whether this high diversity is a characteristic feature of coral reefs or is unique to the Okinawa-Jima ecosystem. Providing more context by comparing with other studies will strengthen the paper's contribution to the broader understanding of coral reef biodiversity.
2.  The introduction mentions the potential of nematodes as bioindicators for coral reefs, but the manuscript does not fully explore this idea. It would be valuable if the authors could elaborate on this by discussing whether any specific nematode species, possibly keystone species, are uniquely found in coral reef habitats, particularly in the Okinawa-Jima region. Such species could potentially be bioindicators of coral reef health or environmental stress. Providing examples of how these species could be used as bioindicators would strengthen this argument and clarify how nematode communities can contribute to coral reef monitoring.
3. As mentioned in the introduction, this study also aims to analyze the nematode phylogeny in these habitats. However, the connection between phylogenetic analysis and community studies is unclear. I suggest the authors include a separate section (another section title?) that explains how increasing 18S and 28S sequences in the database can provide a more accurate phylogenetic topology of nematodes, enhancing the understanding of their evolutionary relationships, but not their role in community structure.
4. It would be clearer to illustrate the differences in nematode genera found in each habitat by providing a Venn diagram for the three habitats, similar to Figure 6.

Experimental design

-

Validity of the findings

While the paper discusses the diversity of nematode communities, it does not provide detailed information on how this diversity was quantified. The authors may consider incorporating established indices, such as Chao1 (which estimates species richness) and Shannon's Diversity Index (which accounts for both species richness and evenness). These indices are widely used in biodiversity studies. They would provide a more robust evaluation of the diversity in the nematode communities across different habitats.

---

## Round 0.2 · Minor Revisions

Please, take the reviewer's minor comments into account.

Reviewer 1 ·

Basic reporting

The manuscript resubmitted includes the changes suggested in the first revision, and the document is now clearly focused on free-living marine nematodes.

Regarding the histograms of genera, I understand that displaying the full diversity in a single graph can be chaotic. However, presenting the data by habitat within each location could offer more meaningful visual insights into the presence or absence of genera across habitats and sites. For clarity, the authors might consider focusing on the 20 most abundant genera, grouping the less abundant ones into a single category labeled 'Other.

In line 93, the authors mention in the objective that this study helps to evaluate the most efficient methods for studying nematode diversity in the future. Numerous studies confirm that morphology and DNA barcoding are complementary, and the use of both approaches results in stronger evidence for identification. The authors could improve the objective by considering the contribution this study provides, which is the generation of new sequences for genera not registered in databases.

Verify the comment in line 325

Verify the comment in line 334

Experimental design

The data presented in Table 2 includes environmental variables that were not included in the statistical analysis; it would be better to present only the information used in the study. Also clarify the temp for depth, temp, Temp. for DO, all the values are in different scales without Units.

The format of both Tables (2 and 3) could be improved if they only left 2 lines at the top and one at the bottom.

Validity of the findings

The addition of the Chao1 and Shannon indices now shows robust information about the diversity and richness of the sites, allowing for a better comparison between habitats

Lines 535-536. This sentence needs to be improved. It is not possible to compare molecular and taxonomic results, on the one hand, because amplification success depends on other factors, and on the other, identification depends on the available database. DNA barcoding is used to reinforce taxonomic identification, to discover cryptic species, new species, or species complexes, but not directly to determine the diversity of a site. The discussion should be directed towards cases such as Desmodora or Cheironchus sp., currently classified within the family Selachinematidae, but in this study, molecular analysis placed it between the families Linhomoeidae and Desmodoridae. More sequences of Cheironchus of different localities will show more evidence that will help clarify their position in the tree.

Annotated reviews are not available for download in order to protect the identity of reviewers who chose to remain anonymous.

Reviewer 2 ·

Basic reporting

The revised version is much improved. I have no further suggestions.

Experimental design

I have no further suggestions.

Validity of the findings

I have no further suggestions.

Additional comments

I have no further suggestions.

---

## Round 0.3 · accepted · Accept

Thank you for addressing all reviewers' comments. I am happy with the current version and recommend it for publication.

Reviewer 1 ·

Basic reporting

The authors have included the changes suggested; I have no further suggestions.

Experimental design

I have no further suggestions.

Validity of the findings

I have no further suggestions.

Additional comments

I have no further suggestions.